# Mesenchymal Meis2 controls whisker development independently from trigeminal sensory innervation

Mehmet Mahsum Kaplan[1], Erika Hudacova[1,2], Miroslav Matejcek[1], Haneen Tuaima[3], Jan Křivánek[3], Ondrej Machon[1,4]*

[1]Department of Developmental Biology, Institute of Experimental Medicine, Czech Academy of Sciences, Prague, Czech Republic; [2]Department of Cell Biology, Faculty of Science, Charles University, Brno, Czech Republic; [3]Department of Histology and Embryology, Faculty of Medicine, Masaryk University, Brno, Czech Republic; [4]Laboratory of Transcriptional Regulation, Institute of Molecular Genetics, Czech Academy of Sciences, Prague, Czech Republic

## eLife Assessment

This study provides **valuable** insight into the role of Meis2 in whisker hair follicle formation and confirms prior work that nerves are dispensable for this process. The **solid** imaging techniques support the authors' conclusions, however the data provides limited evidence to support the mechanism of Meis2 in whisker formation.

*For correspondence:
ondrej.machon@iem.cas.cz

Competing interest: The authors declare that no competing interests exist.

**Abstract** Hair follicle development is initiated by reciprocal molecular interactions between the placode-forming epithelium and the underlying mesenchyme. Cell fate transformation in dermal fibroblasts generates a cell niche for placode induction by activation of signaling pathways WNT, EDA, and FGF in the epithelium. These successive paracrine epithelial signals initiate dermal condensation in the underlying mesenchyme. Although epithelial signaling from the placode to mesenchyme is better described, little is known about primary mesenchymal signals resulting in placode induction. Using genetic approach in mice, we show that *Meis2* expression in cells derived from the neural crest is critical for whisker formation and also for branching of trigeminal nerves. While whisker formation is independent of the trigeminal sensory innervation, MEIS2 in mesenchymal dermal cells orchestrates the initial steps of epithelial placode formation and subsequent dermal condensation. MEIS2 regulates the expression of transcription factor *Foxd1*, which is typical of pre-dermal condensation. However, deletion of *Foxd1* does not affect whisker development. Overall, our data suggest an early role of mesenchymal MEIS2 during whisker formation and provide evidence that whiskers can normally develop in the absence of sensory innervation or *Foxd1* expression.

## Introduction

The local cell environment provides essential cues that determine cell fate decisions, proliferation, and migration. The molecular communication between the developing epithelium and the underlying cell environment is crucial for the formation of multiple structures, which are derived from the embryonic epithelium, such as the eye lens, hair follicles (HFs) including whisker (vibrissae) follicles, teeth, taste papillae, the intestine lumen, or salivary glands. In the developing vertebrate head, these tissue cross-talks employ neural crest-derived mesenchyme with the epithelium, myogenic progenitors, or

cartilage. Critical steps in whisker follicle (WF) development are very similar to normal HF development (*Wrenn and Wessells, 1984*), which has been studied much more extensively. HF morphogenesis is initiated by thickening of the epithelium forming a placode (Pc), which subsequently induces condensation of underlying fibroblasts (Fb) termed dermal condensation (*Sennett and Rendl, 2012*). Beta-catenin-dependent WNT signaling is essential for Pc induction (*Huelsken et al., 2001*), which initiates the EDA pathway (*Zhang et al., 2009*), FGF20 (*Huh et al., 2013*), and SHH signaling (*Botchkarev et al., 1998*) in the epithelium. Abrogation of any of these signaling pathways leads to the arrest of HF development. However, the succession of initial inductive steps in HF formation is still unclear. Pc formation requires hitherto unknown primary signals from dermal Fb at the pre-dermal condensate (DC) stage that is characterized by expression of *Foxd1* (*Mok et al., 2019*) and *Prdm1* (*Manti et al., 2022*). Recent studies employing single-cell transcriptomics helped elucidate the molecular determination of Pc and DC induction. *Mok et al., 2019*, described a wave of transcription factors that define cell fate trajectories, starting in dermal Fb underlying the epithelium before Pc induction (pre-DC stage), toward DC and the late DC stage when Pc invagination commences. For instance, expression of *Foxd1* transcription factor was detected in pre-DC Fb that are also characterized by expression of *Twist2*, *Lef1*, and *Smad3* (*Mok et al., 2019*; *Sennett et al., 2015*). DC cells predominantly express *Sox2*, *Foxd1*, *Ptch1*, and *Bmp4* (*Gupta et al., 2019*; *Ge et al., 2020*). On the other hand, DC initiation requires canonical WNT and FGF20 signaling in Pc, suggesting a reciprocal molecular cross-talk between the dermal mesenchyme and overlying epithelium (*Mok et al., 2019*; *Gupta et al., 2019*). Transplantation experiments suggested that mesenchymal cells modulate WNT signaling in Pc by activation of RSPO1; however, RSPO1 is not sufficient for HF induction even in combination with BMP inhibition (*Mäkelä and Mikkola, 2023*). Although these data comprehensively map molecular events during HF initiation, identification of the primary inductive signal for Pc induction coming from dermal Fb is still missing.

Epithelial structures such as whiskers or teeth are formed at precisely defined locations. Whiskers are arranged in an accurate rectangular pattern in five rows, each containing four to nine follicles. Each whisker is innervated by a separate whisker-to-barrel circuit that is reflected in a clear topographic organization of the barrel cortex (*Yang et al., 2018*). Axons of the trigeminal (TG) nerve ending in the infraorbital nerve innervate each whisker germ in the maxillary prominence. It has been suggested that early innervation does not play a role in determining WF pattern as distribution of TG axons is widespread and random in the maxillary process, and no one-to-one spatial correlation between nerves and WFs was found (*Wrenn and Wessells, 1984*; *Davies and Lumsden, 1986*). This view has been challenged by the observation of a regular pattern of the nerve plexus in the snout half a day prior to the pattern of WFs (*Van Exan and Hardy, 1980*; *Maklad et al., 2010*). However, these studies do not imply whether WFs develop normally in vivo in the complete absence of innervation. Although normal vessel ring organization in the WFs of mice lacking TG sensory innervation has been reported (*Oh and Gu, 2013*), a comprehensive description of the WF development was not studied in this context. Therefore, while existing literature points to a scenario with an unlikely role of innervation for the WF formation and patterning (*Andrés and Van der Loos, 1982*; *Andrés and Van Der Loos, 1983*; *Lillesaar and Hildebrand, 1999*), a more direct and conclusive in vivo study is needed to resolve this issue.

Similarly, the influence of the TG nerves for tooth development has not been conclusively demonstrated. Organ culture methods showed no involvement of TG sensory innervation (*Lumsden and Buchanan, 1986*), and tooth germs can trigger their own innervation (*Erdélyi et al., 1987*). Along these lines, diastema tooth primordia develop and subsequently disappear with no TG innervation (*Løes et al., 2002*). However, the existence of earlier pioneering axons in the TG system has been reported; therefore, tooth development might be initiated by interaction with the early arriving pioneering axons (*Stainier and Gilbert, 1990*). Moreover, roles of neurogenic factors have been shown during tooth development (*Sandor et al., 2014*; *Tachibana et al., 2014*; *Moiseiwitsch and Lauder, 1996*). A surgical removal of (TG) axons innervating teeth led to tooth degeneration (*Zhao et al., 2014*; *Adameyko and Fried, 2016*). Hence, cranial nerves might deliver morphogenetic signals that orchestrate tooth maintenance. The role of innervation during tooth development was also supported by *Kaukua et al., 2014*, who showed that dental mesenchymal stem cells are recruited from the nerve-associated glial cell. Taken together, it has been suggested that innervation play a role in tooth development (*Duan et al., 2022*). However, in a model system where innervation is completely missing, tooth development has not been analyzed.

Thus, peripheral nerves and accompanying glia contribute to the mesenchymal tooth germ niche. Nonetheless, it is still unclear what cellular mechanisms regulate precise positions of whiskers in the snout. Turing reaction-diffusion mathematical model can be applied for oscillating tissue patterning that is orchestrated by diffusion of short-range activating morphogen that triggers expression of long-range inhibitor. Reaction-diffusion-based instruction of the epithelium by combinatorial WNT, FGF, and BMP signaling is capable of HF placode induction in a highly restricted regular pattern (*Glover et al., 2017*). However, a highly specific rectangular arrangement and relatively long distance among WFs within the developing snout may indicate that other molecular determinants play a role in spatio-temporal control of WF induction. This revives the previous speculations that an axonal network may be involved in WF positioning. In this report, we assessed the scenario that the neural network is implicated in the control of accurate geometrical arrangement of WFs.

Here, we generated *Meis2* conditional mutants using the Wnt1-Cre2 driver (*Lewis et al., 2013*) targeting exclusively the neural crest-derived tissues, including the dermal mesenchyme and cranial nerves, without detectable recombination in the overlying epithelium (*Fabik et al., 2020*). Although *Meis2* expression in the epithelium of Wnt1-Cre2; *Meis2* <sup>fl/fl</sup> (*Meis2* cKO) mutants was preserved, whisker development was arrested at the Pc induction stage. This is documented by failure in induction of expression of placodal genes such as *Shh*, *Edar,* and *Lef1*. We further show that severely affected branching of TG nerves innervating WFs cannot cause their developmental arrest because WFs form normally even in a complete absence of TG nerves. Expression of *Meis2* in dermal Fb is therefore necessary for induction of whisker placodes followed by dermal condensation. Protein expression analysis suggests that MEIS2 transcription factor is essential for triggering a primary inductive signal from the mesenchyme to the epithelium to initiate WF placode formation. While single-cell transcriptomics and immunohistochemical validations showed that *Meis2* operates upstream of *Foxd1*, primary pre-DC marker, we report a dispensable role of *Foxd1* during WF development. Overall, our data not only describe an essential role of mesenchymal *Meis2* for WF development but also document normal WF formation even in the complete absence of innervating axons.

## Results

### *Meis2* expression in cranial neural crest-derived cells is required for correct TG nerve projections and whisker development

To study whisker morphogenesis, we used Wnt1-Cre2; *Meis2*<sup>fl/fl</sup> conditional mutants, in which *Meis2* is deleted in neural crest-derived cells (hereafter, *Meis2* cKO). Micro-computed tomography (micro-CT) of embryonic day 15.5 (E15.5) embryos revealed severely compromised development of whiskers (*Figure 1A*). Missing whiskers were also observed whole-mount embryonic heads stained with placodal marker SOX9 antibody after light-sheet microscopy scanning at as early as E13.5 (*Figure 1B*). It is noteworthy that we observed in some cases a small number of SOX9+ WFs in *Meis2* cKO. They most likely represented normal follicles that 'escaped' from Wnt1-Cre2-mediated deletion of *Meis2*. The number of WF 'escapers' in *Meis2* cKO snouts varied among samples. Some mutant embryos completely missed WFs, but some showed a reduction in their numbers. Overall, quantification of the WF number documented a reduction to 5.7 ± 2.0% at E12.5 and to 17.1 ± 5.9% at E13.5 in the mutants (n=at least 3 embryos for E12.5, n=at least 8 embryos for E13.5, mean ± sem) (*Figure 1—figure supplement 1A*). Analysis of placode thickness/follicle length and DC area revealed that these escaper whiskers have normal morphology (*Figure 1—figure supplement 1B and C*). To analyze expression of *Meis2* in the snout, we performed fluorescent immunohistochemistry. MEIS2 protein was detected in the epithelium, in the most superficial dermis including DC. In *Meis2* cKO, mesenchymal expression of *Meis2* virtually disappeared (*Figure 1C*, asterisk, *Figure 1—figure supplement 1D*) while its epithelial expression was maintained (*Figure 1C*, arrowheads, *Figure 1—figure supplement 1D*), which corresponds to tissue-specific Cre recombination of Wnt1-Cre2 (*Lewis et al., 2013*; *Fabik et al., 2020*). It is interesting that MEIS2 signal was also remarkably reduced around sole WF 'escapers' (*Figure 1C*, arrow) but we cannot exclude that, due to incomplete Cre-mediated deletion, a minor expression of *Meis2* below the immunohistochemical detectability threshold is sufficient for WF induction. To verify the tissue specificity in the snout region, we crossed Wnt1-Cre2 mice with mTmG mouse strain, where only cells derived from the neural crest express membrane-localized GFP. Whole-mount immunofluorescence imaging confirmed the Wnt1-Cre2 activity in neural crest derivatives in

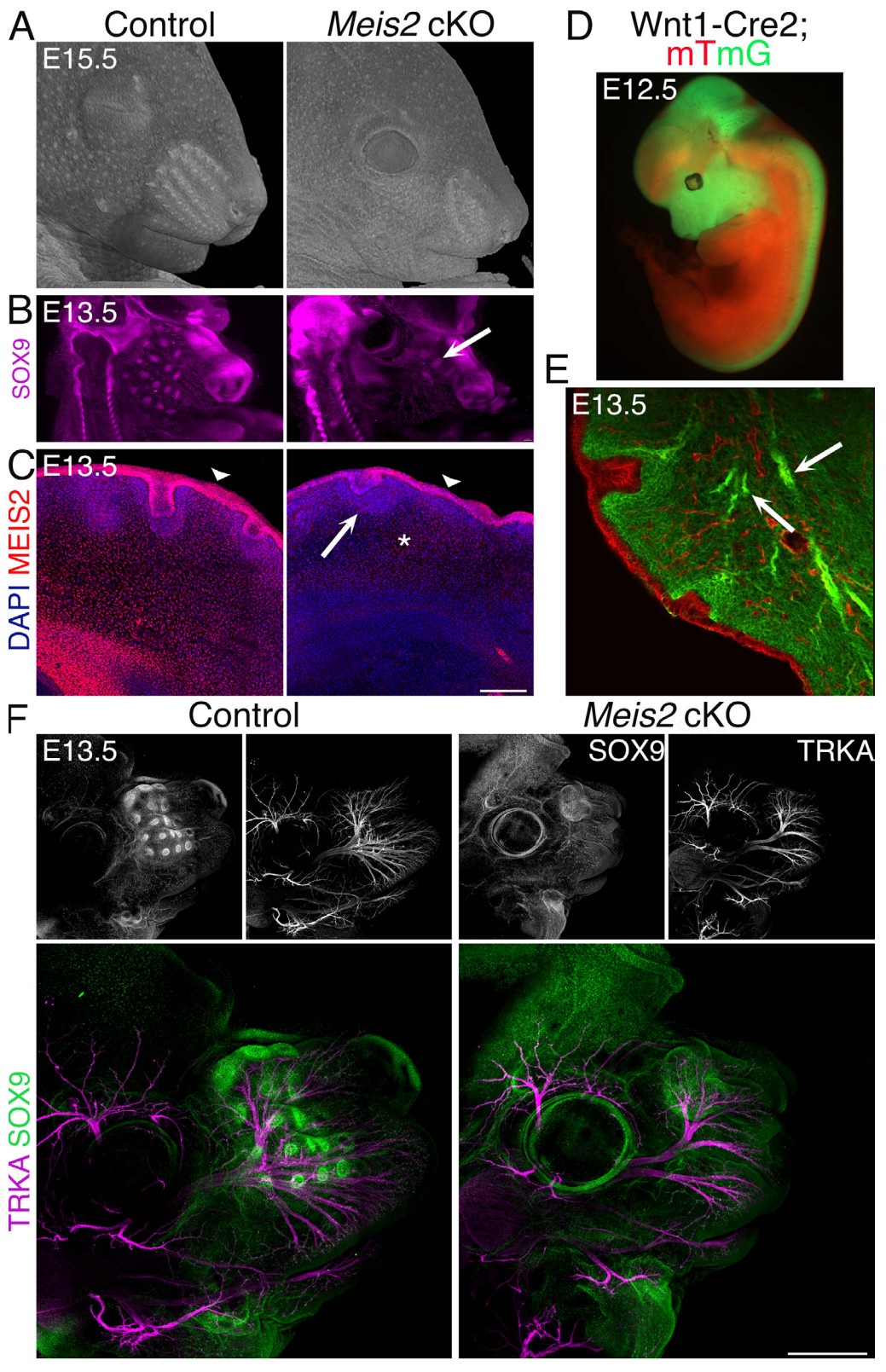

**Figure 1.** Severe whisker follicle (WF) development in *Meis2* cKO embryos. (**A**) Micro-computed tomography (micro-CT) images of control and *Meis2* cKO embryos at embryonic day 15.5 (E15.5) showing aberrant whisker phenotype in *Meis2* cKO mice. (**B**) Light-sheet microscopy of Sox9 whole-mount immunostaining of WF. Arrow indicates example of an 'escaper' whisker. (**C**) MEIS2 immunofluorescence on frontal frozen sections of E13.5

*Figure 1 continued on next page*

*Figure 1 continued*

snouts showing MEIS2 expression in the dermis, dermal condensate (DC), and epithelium including placode (Pc). Arrowheads indicate epithelial expression of MEIS2 in both control and *Meis2* cKO mice. Arrow and asterisk show disappearing MEIS2 expression in the DC and dermis, respectively, in the mutant. Scale bar: 150 µm. (**D**) Wnt1-Cre; *mTmG* embryos showing Cre recombination specificity in the craniofacial area, midbrain, and dorsal spinal cord. Recombined cells are labeled by membrane-localized GFP in green, and non-recombined cells labeled by membrane-localized tdTomato in red. (**E**) Frontal sections of Wnt1-Cre; *mTmG* snouts documenting Cre recombination in the neural crest-derived mesenchyme, cranial nerve projections without recombination in the overlying epithelium. Arrows show trigeminal (TG) nerve branches. (**F**) Whole-mount staining of *Meis2* cKO heads at E13.5 with SOX9 and TRKA antibodies showing almost absence of WFs and compromised branching of the TG nerve in mutants. Scale bar: 500 µm.

The online version of this article includes the following figure supplement(s) for figure 1:

**Figure supplement 1.** Normal sizes of escaper whisker follicles (WFs) and trigeminal (TGs) in *Meis2* cKO mice.

**Figure supplement 2.** Whisker phenotype persists at embryonic day 18.5 (E18.5) *Meis2* cKO mice.

---

the craniofacial area, midbrain, and dorsal spinal cord (*Figure 1D*). In the snout, Cre activity was restricted to dermal mesenchyme without a detectable recombination in the overlying epithelium (*Figure 1E*). Moreover, branches of TG sensory nerves were labeled with GFP (*Figure 1E*, arrows), consistent with the previously known origin of TG neurons mainly from neural crest cells (*Leonard et al., 2022*). Whiskers are innervated by the axons of the infraorbital nerve, a branch of the TG nerve. To examine the axonal outgrowths associated with whisker germs, we co-labeled the HF marker SOX9 with the sensory nerve marker TRKA in whole-mount heads at E13.5. *Figure 1F* and *Videos 1 and 2* show that missing whiskers in mice lacking *Meis2* in neural crest-derived cells were accompanied by severely affected TG nerve growth and branching. Because defects in nerve branching were not limited to the infraorbital nerve and were also observed in other TG branches, this phenotype is most likely not caused by missing WFs themselves. Moreover, the size of the TG ganglion and the nerve exit from the ganglion appeared grossly normal (*Figure 1—figure supplement 1E*). These observations suggest that defects in the TG nerve morphology occur during branching and fasciculation of axonal projections through the craniofacial region. Overall, these results indicate an essential role of *Meis2* not only in the development of the WFs but also for the branching of TG nerves that innervate them.

Normal sizes of the escaped WFs in *Meis2* cKO mice at E12.5 and E13.5 (*Figure 1—figure supplement 1B and C*) suggest that WF phenotype in the mutants is not due to a delay in their development since in this case WFs would appear smaller. In order to further exclude this possibility that WF

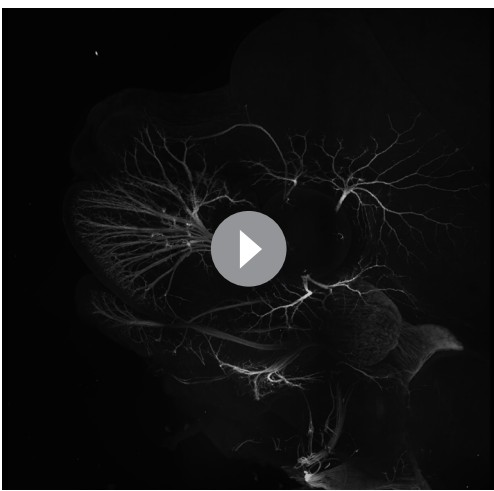

**Video 1.** Embryonic day 13.5 (E13.5) whole-mount half head of a control mouse stained with TRKA antibody and scanned with a spinning disk microscope at 1 µm z-step size.

https://elifesciences.org/articles/100854/figures#video1

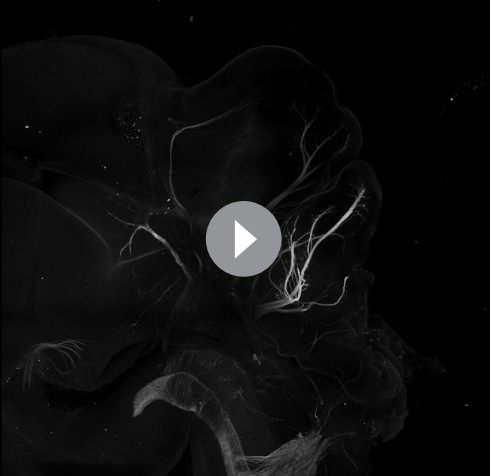

**Video 2.** Embryonic day 13.5 (E13.5) whole-mount half head of a *Meis2* cKO mouse stained with TRKA antibody and scanned with a spinning disk microscope at 1 µm z-step size. Compared to its control littermate in *Video 1*, peripheral trigeminal (TG) nerve branching is considerably defected in the mutant.

https://elifesciences.org/articles/100854/figures#video2

formation is delayed in *Meis2* cKO mice, we analyzed whisker phenotype at later embryonic stages. Sections of E15.5 micro-CT images showed aberrant WF phenotype (*Figure 1—figure supplement 2A*). Also, at E18.5, the pattern of WFs were missing in the mutant (*Figure 1—figure supplement 2B*). Staining of E18.5 snout section with placodal marker EDAR antibody revealed that, similarly to E13.5, a small number of escaped WFs in *Meis2* cKO with comparable depths to control WFs formed (*Figure 1—figure supplement 2C*, arrows). However, this aberrant whisker phenotype persisted until E18.5. These data indicate that the whisker phenotype in the mutants is not due to a developmental delay but rather suggest a direct involvement of mesenchymal *Meis2* in mechanisms initiating WF formation. Interestingly, at E18.5 stage, HFs became detectable in the snout with a similar appearance in the mutants, suggesting a specific role of *Meis2* in WFs. This could be due to an early role of *Meis2* in the mesenchyme because HFs develop later.

## TG innervation is not required for WF development

Aberrant whisker development in *Meis2* cKO could be explained by an indispensable function of *Meis2* in mesenchymal cells or by insufficient TG sensory innervation. The latter scenario takes into account the possibility that the nerve or nerve-associated Schwann cells produce factors that are necessary for WF formation. Thus, compromised branching of sensory axons in the *Meis2* cKO snout might result in insufficient secretion of crucial WF-inducing factors. In order to directly assess the function of sensory innervation in WF development, we analyzed *Neurog1*[-/-] mice, which do not develop some cranial nerves, including TG nerves, and therefore lack sensory innervation of the developing whiskers (*Ma et al., 1998*). Immunostaining of E13.5 100-μm-thick snout cryosections for placodal marker SOX9, axonal marker TUJ1, and sensory nerve marker TRKA antibodies revealed that whiskers developed normally in *Neurog1*[-/-] mutants even in the complete absence of innervating sensory axons (*Figure 2A*). Lack of any TUJ1-labeled axon fascicules in the proximity of the normally developed WFs in the *Neurog1*[-/-] snout further excludes the possibility of non-sensory neurons triggering WF development in the absence of sensory innervation. In order to assess the role of sensory innervation for overall whisker patterning, we stained E12.5 and E13.5 snout whole mounts with SOX9 and EDAR antibodies, marking initial stages of WF development (*Zhang et al., 2009*), i.e., Pc formation. These results revealed essentially identical whisker patterning between control and *Neurog1*[-/-] mice (*Figure 2B*). EDAR staining performed at E12.5 10 μm FFPE sections further confirmed normal placodal expression of EDAR, indicating normal initiation of the WF development (*Figure 2C*). WF formation was also validated by micro-CT at E17.5, showing normal progression of WF development at later stages (*Figure 2D*, top) even when TG ganglia were absent (*Figure 2D*, bottom, arrows). Overall, these data unambiguously show that whisker formation does not require TG sensory innervation and exclude the possibility that impaired nerves or nerve-associated cells in *Meis2* cKO resulted in impaired WF induction. Therefore, it can be confidently concluded that the dermal mesenchyme expressing *Meis2* is evidently critical for WF development.

In addition to whisker phenotype, we analyzed tooth morphology in *Neurog1*[-/-] mice since the role of innervation for tooth development represents a long-standing controversy and has not been directly tested previously (*Lumsden and Buchanan, 1986*; *Løes et al., 2002*; *Stainier and Gilbert, 1990*; *Duan et al., 2022*; *Maeda et al., 2019*; *Weil et al., 1995*). Interestingly, we did not detect any defects in tooth development (*Figure 2—figure supplement 1A*) which was corroborated also by micro-CT images from E17.5 mice (*Figure 2—figure supplement 1B*), reporting an innervation-independent formation of teeth in *Neurog1*[-/-] mice.

## Mesenchymal *Meis2* operates upstream of epithelial EDAR signaling during initial steps of WF formation

Our data strongly suggest that *Meis2* expression in the mesenchyme is required for induction of placodal expression of *Sox9* in WF. Since a number of regulators of HF development were identified to function upstream of placodal *Sox9* expression, we tested these successive molecular events leading to WF induction. SHH release in basal placodal cells was shown to trigger *Sox9* expression in suprabasal placodal cells. This is achieved by asymmetric cell division resulting in suprabasal daughter cells with low WNT activity so that SHH can drive *Sox9* expression (*Ouspenskaia et al., 2016*). We performed RNA in situ hybridization to evaluate *Shh* expression in *Meis2* cKO mice and control littermates at E13.5. These data revealed a remarkably decreased number of *Shh*-positive foci in *Meis2*

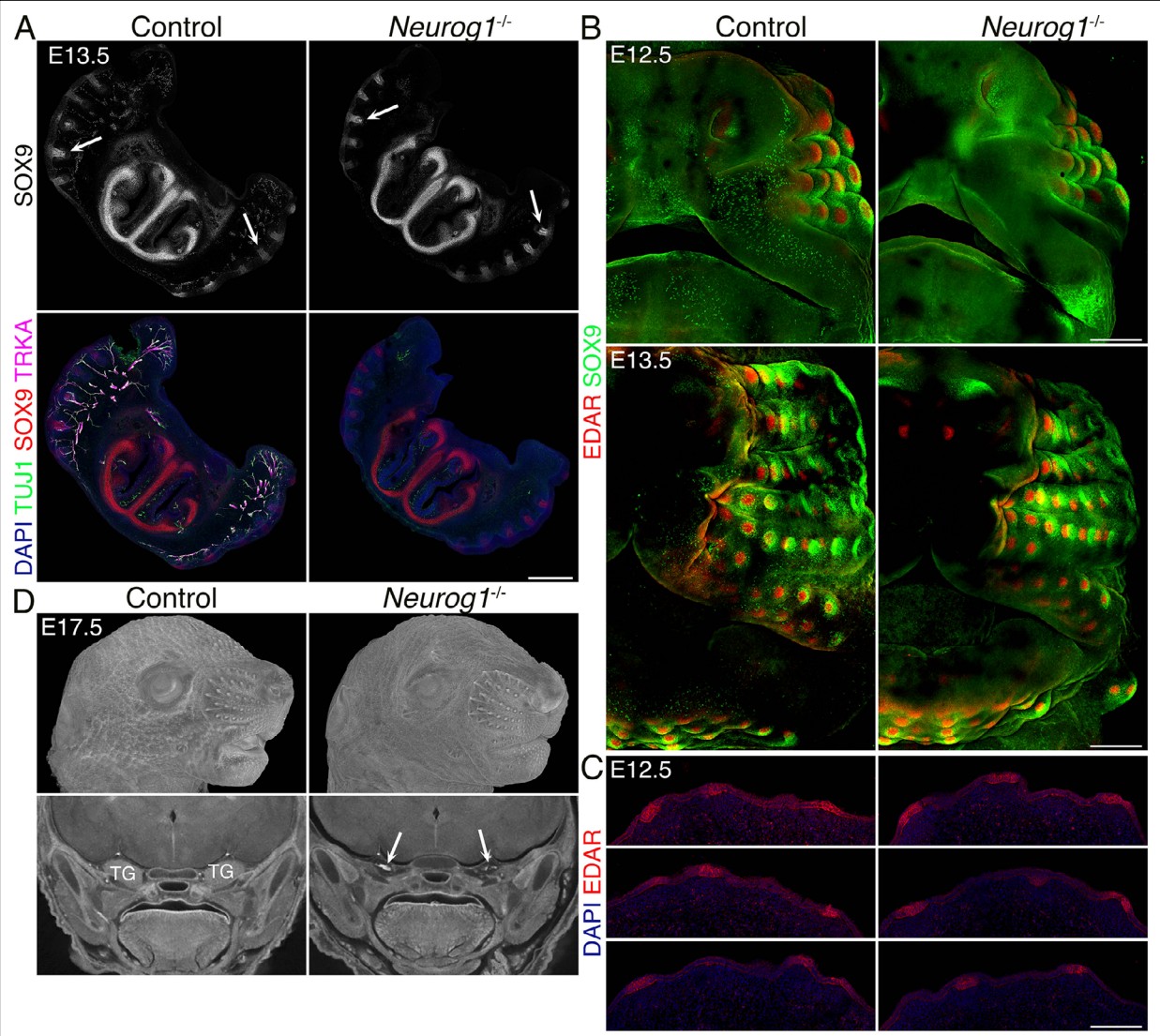

**Figure 2.** *Neurog1⁻ᐟ⁻* (KO) embryos lack the trigeminal (TG) nerve, but whisker follicle (WFs) development is normal. (**A**) Triple immunostaining of 100 μm sections shows the absence of TG nerve projections (TUJ1+, TRKA+) and normal WF (SOX9+) in *Neurog1⁻ᐟ⁻* mice. Arrows in black and white images show examples of invagination of whisker placodes labeled by SOX9 antibody. Scale bar: 500 μm. (**B**) Whole-mount immunostaining of WFs with SOX9 and EDAR antibodies showing normal WF morphology and patterning in mutants at embryonic day 12.5 (E12.5) (top) and 13.5 (bottom). Scale bars: 300 μm. (**C**) EDAR staining of 10 μm sections at E12.5 showing normal initiation of placode formation in *Neurog1* mutants. Scale bar: 150 μm. (**D**) Micro-computed tomography (micro-CT) images reveal normally developed whiskers (top) at E17.5 in mutants while TGs are lacking (bottom, arrows).

The online version of this article includes the following figure supplement(s) for figure 2:

**Figure supplement 1.** Normal tooth development in *Neurog1⁻ᐟ⁻* mice.

cKO mice (*Figure 3A*, arrow), showing that placodal expression of *Shh* is regulated by mesenchymal MEIS2 protein. Again, we observed a small number of *Shh*-positive WF escapers (*Figure 3A*, arrow). EDA/NFκB signaling in Pc has been shown to act upstream of Shh signaling before Pc invagination (*Zhang et al., 2009*). We therefore stained whole-mount E12.5 snouts from *Meis2* cKO mice and controls with SOX9 and EDAR antibodies. As seen in *Figure 3B*, EDAR protein was localized to epithelial WF Pc and overlapped with SOX9 staining in controls. In contrast, in *Meis2* cKO mutants, both EDAR and SOX9 were hardly detectable, documenting that *Meis2* mesenchymal function is upstream of EDA signaling in Pc. Whole-mount EDAR staining was confirmed on tissue sections which detected significantly reduced number of EDAR-positive foci in the *Meis2* cKO epithelium. Moreover, EDAR-negative regions within the mutant epithelium did not show any signs of morphological Pc formation as inspected by DAPI staining (*Figure 3*, *Figure 4—figure supplement 1A*). Overall, these

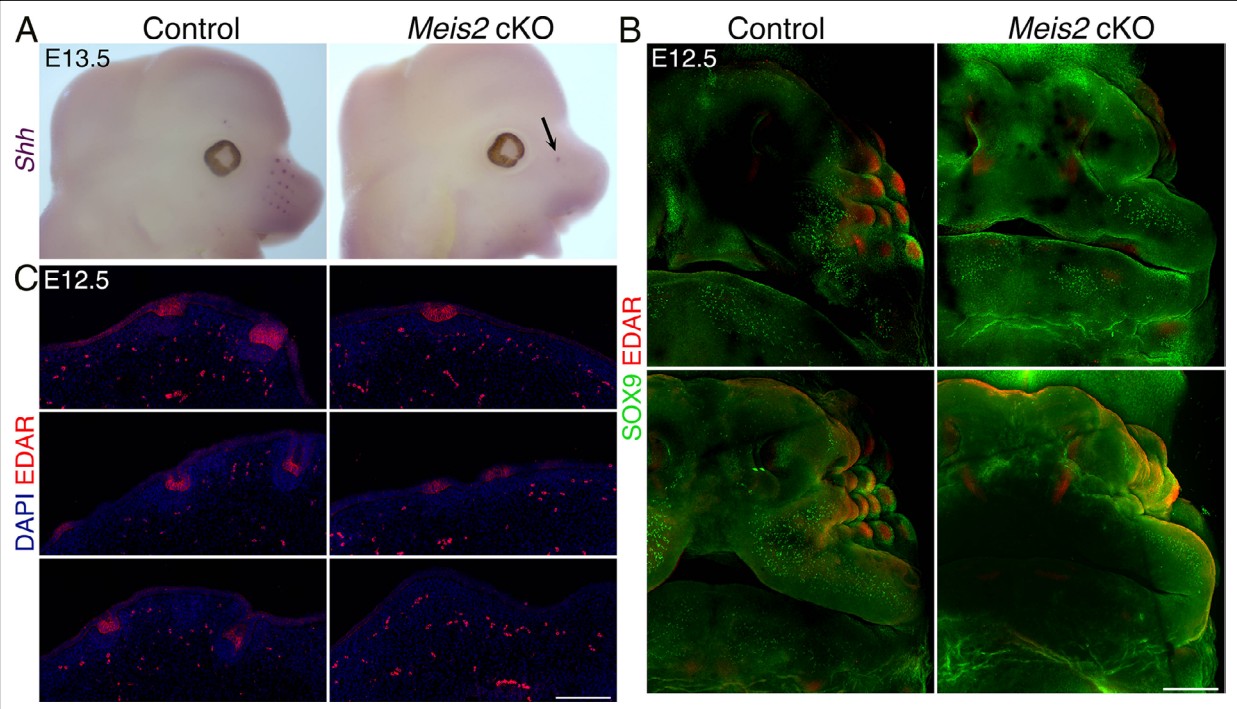

**Figure 3.** Induction and progression of whisker follicle (WF) development is compromised in *Meis2* cKO. (**A**) Whole-mount in situ hybridization of *Shh* mRNA documenting loss of WFs in mutants. Arrow shows an escaper whisker. (**B**) Whole-mount immunostaining of WFs with SOX9 and EDAR antibodies showing absence of WFs in *Meis2* cKO at embryonic day 12.5 (E12.5). Two examples for each genotype are shown. Scale bar: 300 μm. (**C**) EDAR staining of 10 μm sections showing placode formation arrest in mutants. Scale bar: 150 μm.

data demonstrate that mesenchymal *Meis2* operates at very early stages of WF formation in which it participates in forming a primary inductive mesenchymal signal for EDAR+ Pc formation. This early arrest in mutants affects all subsequent steps of WF development including *Shh* and *Sox9* expression and Pc and DC formation.

### *Meis2* deletion in NCC-derived cells affects epithelial but not dermal WNT signaling

It has been shown that WNT/β-catenin signaling acts upstream of EDA/NFκB signaling as the earliest molecular pathway identified during HF formation (*Zhang et al., 2009*). Epithelial WNT activity is required for activation of dermal canonical WNT signaling, while dermal WNT signaling reciprocally affects patterned WNT activity in the epithelium (*Chen et al., 2012*). We therefore assessed canonical WNT signaling in *Meis2* cKO snouts by several types of WNT signaling readouts. First, we used LEF1 immunofluorescence because its expression been widely used as proxy for WNT signaling in HF (*DasGupta and Fuchs, 1999*; *Matos et al., 2020*) and *Lef1* is also required for whisker formation (*Kratochwil et al., 1996*). *Matos et al., 2020*, showed that dermal cells expressed *Lef1* broadly whereas in the epithelium, *Lef1* expression was concentrated in Pc cells compared to interfollicular regions. During Pc downgrowth in HF, basal Pc cells displayed LEF1 signal similarly to Shh and EDAR, in contrast to suprabasal placodal cells lacking LEF1. This is consistent with the view that high WNT activity is seen in the basal cells whereas suprabasal cells are WNT low (*Matos et al., 2020*). In WF, we similarly detected robust LEF1 signal in Pc that was remarkably lower in interfollicular regions (*Figure 4A*, *Figure 4—figure supplement 1B and C*). However, in distal snout regions with no WFs, the epithelium showed confluent LEF1 signal (*Figure 1—figure supplement 1C*), suggesting that LEF1 signal is regionally dependent Pc indicator. In contrast to the epithelium, DC underneath invaginating Pc consistently displayed decreased LEF1 staining compared to peri-DC neighboring regions (*Figure 4A*, *Figure 4—figure supplement 1B*). In *Meis2* cKO, we rarely detected a patterned placodal increase in LEF1 staining in the expected sites of WF positions (*Figure 4A*, *Figure 4—figure supplement 1C*). Quantification of LEF1-stained E12.5 10 μm FFPE sections revealed a significant decrease

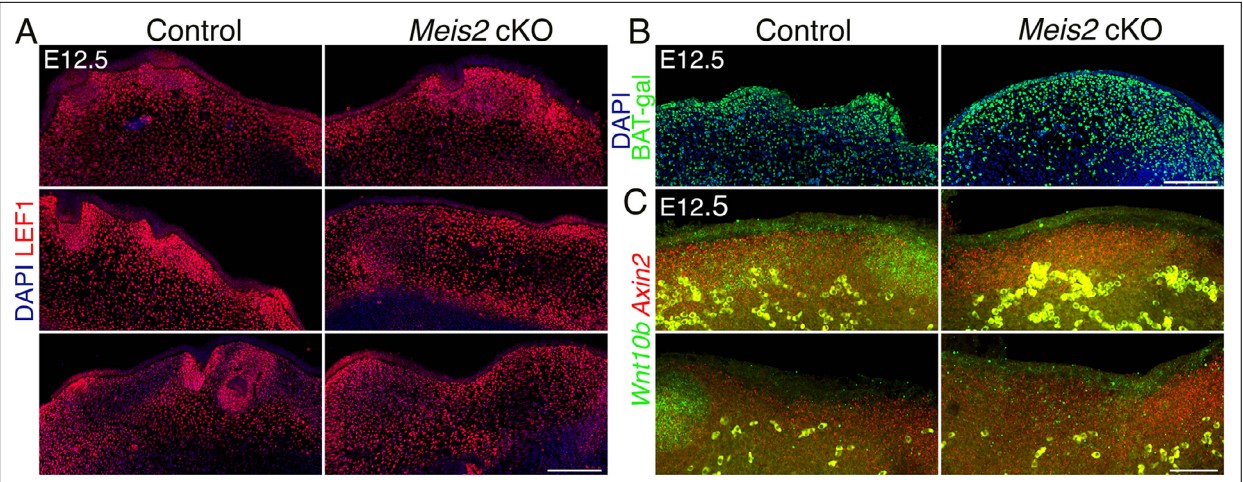

**Figure 4.** WNT signaling in the epithelium is affected by deletion of *Meis2* in the dermal mesenchyme. (**A**) LEF1 staining of 10 μm paraffin frontal sections of snouts showing abundance of LEF1 in the dermal mesenchyme, while in the epithelium, it is concentrated in placodes in regions of whisker follicle (WF) appearance. Similar to missing placodes, LEF1 is also lost in *Meis2* cKO epithelium. Scale bars: 150 μm. (**B**) Beta-galactosidase immunostaining of sections from WNT reporter BAT-gal controls and *Meis2* cKO; BAT-gal mutants showing widespread galactosidase signal in the dermis and epithelium. Scale bars: 150 μm. (**C**) In situ hybridization HCR-FISH using probes for *Axin2* and *Wnt10b*. In the epithelium, *Wnt10b* is detected in placodes, and the signal is lost in *Meis2* cKO. Similarly, upregulation of *Axin2* is not observed in the expected WF loci. In the dermal mesenchyme, *Axin2* mRNA is detected throughout the dermis regardless of WFs. This pattern was not changed in *Meis2* cKO. Scale bar: 50 μm.

The online version of this article includes the following figure supplement(s) for figure 4:

**Figure supplement 1.** EDAR and LEF1 expression in *Meis2* cKO mice.

in LEF1+ placodal sites per section from 2.89±0.39 in control to 1±0.21 *Meis2* cKO (mean ± sem, p=0.0002, two-tailed t-test, n=at least 3 mice, 18 sections) (*Figure 4—figure supplement 1D*, left). This shows that failure of WF initiation in *Meis2* cKO is accompanied by missing expression of *Lef1* in Pc, indicating that canonical WNT signaling is not induced in the Pc epithelium. However, average LEF1 intensity in the escaped Pc of the mutants did not significantly differ from that of control (95.08 ± 6.47 %, p=0.51, two-tailed t-test) (*Figure 4—figure supplement 1D*). On the other hand, LEF1 staining in *Meis2* cKO dermal Fb was confluent and without downregulation in presumptive DC sites, which again confirms early developmental arrest in mutant WFs. Analysis of LEF1 staining intensity in both DC (93.60 ± 6.77%, p=0.305, two-tailed t-test, n=3 mice and 43 DCs for control and 17 DCs for mutants) and non-DC upper dermis (89.91 ± 13.19%, p=0.117, two-tailed t-test, n=3 mice and 37 regions for control and 33 regions for mutants) again did not reveal any significant difference between control and mutant snouts (*Figure 4—figure supplement 1D*). These data indicate that *Meis2* in the mesenchyme does not affect dermal WNT signaling but is required for induction of epithelial WNT activity during Pc formation.

Second, in order to verify WNT activity dynamics in developing WF, we employed WNT reporter mouse strain BAT-gal (*Maretto et al., 2003*) in which multiple TCF-binding sites promote expression of *beta-galactosidase* gene. We crossed this reporter mouse strain to *Meis2* cKO and stained snout frontal sections with an anti-galactosidase antibody. In contrast to LEF1 staining, BAT-gal reporter staining did not show consistent upregulation and concentration of WNT activity in Pc and no corresponding downregulation in DC in controls. It rather exhibited confluent expression in the dermal mesenchyme (*Figure 4B*) that was not changed in mutants. Thus, BAT-gal reporter therefore did not serve as a reliable WNT readout in developing WF in our hands, probably due to the high stability of galactosidase protein compared to RNA (*Machon et al., 2007*), which limited the analysis of the dynamic nature of WNT activity during WF development.

Third, we tested the expression of selected WNT ligands in *Meis2* cKO snouts. *Wnt10b* expression is strongly activated in HF Pc (*Reddy et al., 2001*) and it is also important for HF progression (*Wu et al., 2020*). Thus, *WNT10b* seems the most likely *Wnt* gene activating canonical pathway in HF which is routinely monitored by *Axin2* expression in many tissue contexts including HF (*Gupta et al., 2019*; *Ouspenskaia et al., 2016*). We performed HCR FISH to assess expression of *Wnt10b* and WNT

target gene *Axin2* in snouts at E12.5. Similarly to LEF1 staining, we observed *Axin2* increase in wild-type Pc, while DC and dermal mesenchyme displayed uniform *Axin2* expression, which did not accord with decreased LEF1 in DC (*Figure 4C*, *Figure 4—figure supplement 1B and E*). This might reflect differential responses of *Lef1* and *Axin2* expression to WNT activity in DC. In *Meis2* cKO, upregulation of *Axin2* in the epithelium was not detected which most likely reflected missing Pc. On the other hand, mutant dermal mesenchyme displayed broad *Axin2* expression which was similar to wild-type controls. *Wnt10b* transcripts were detected in normal Pc in controls while this expression disappeared in missing Pc in mutants (*Figure 4C*). Overall, these data suggest that mesenchymal *Meis2* does not affect WNT signaling in the dermis, whereas it is critical for the patterned WNT upregulation in the epithelium. Since WNT activation in the epithelium is one of the earliest steps of HF and WF development, this leads to loss of EDAR and *Shh* markers in *Meis2* cKO WF, and it confirms a very early role of mesenchymal *Meis2* during whisker development.

## *Meis2* regulates dermal cell proliferation and the transition of Fb cell fate to DC

We have recently reported single-cell RNA sequencing (scRNA-seq) datasets for mesenchymal cells derived from the cranial neural crest of E12.5 and E13.5 *Meis2* cKO embryos and their control littermates (*Hudacova et al., 2025*). Cell cluster analysis identified one cluster representing dermal Fb. We subset and further re-clustered using the standard Seurat workflow (in Materials and methods). We identified six clusters, where cluster 2 was defined as DC by cell markers *Sox2*, *Sox18*, *Tbx18*, *Bmp4*, *Lef1*, and *Cdkn1a* (*Figure 5A and B*, *Figure 5—figure supplement 1A*, *Supplementary file 1*). The other clusters were determined to represent Fb annotated by their markers in the heatmap and module scores generated for Fb, pre-DC, DC1, and DC2 markers (https://github.com/kaplanmm/whisker_scRNA, copy archived at *Kaplan, 2025*; *Mok et al., 2019*; *Sennett et al., 2015*; *Figure 5—figure supplement 1B*, *Supplementary file 1*). Cluster 2 displayed low expression of Fb module score genes and high expression of DC2 module score genes, confirming again that cluster 2 represents developed DCs. Transcriptomic analysis showed that the cell number proportion in cluster 2 was substantially lower in *Meis2* cKO mice compared to control littermates (*Figure 5C*), reflecting the WF phenotype in the mutant. Interestingly, cluster 0 also contained a lower cell proportion in the *Meis2* cKO dataset. On the other hand, the cell number proportion in cluster 1 increased in *Meis2* cKO (*Figure 5C*). However, the cell identity of cluster 1 was unclear because it was marked by only 27 genes (*Supplementary file 1*). Cluster 0 revealed GO terms related to cell cycle and chromosome segregation/organization according to GO analysis by clusterProfiler R package (*Wu et al., 2021*) using top 100 markers (*Supplementary file 2*). Therefore, we conclude that cluster 0 very likely represents dividing cells. Lower cell number proportion in cluster 0 in *Meis2* cKO suggested a *Meis2*-dependent regulation of dermal cell proliferation. This hypothesis was tested by performing EdU chase experiments where EdU was injected at E12.5 and embryos were harvested after 2 or 18 hr. Indeed, the 2D area coverage of DAPI+ region by EdU+ region in the upper dermis of *Meis2* cKO snout decreased from 52.96±2.05% to 42.38±2.77% (mean ± sem, t-test p=0.0057) in 2 hr pulse and from 52.17±0.81% to 45.76 ± 1.85% (mean ± sem, t-test p=0.0052) in 18 hr pulse (*Figure 5D and E*). These EdU experiments confirmed scRNA-seq data and indicated a *Meis2* function in proliferation of upper dermal cells which might contribute to the aberrant whisker phenotype of the mutant.

Because cell migration is a critical process for the migration of Fb cells into DC during cell fate acquisition (*Biggs et al., 2018*), we plotted cell adhesion and ECM module scores, both of which were found to be increased in mutants (*Figure 5F and G*). Thus, such an increase in cell interactions might limit Fb migration efficiency by decreasing conductivity of the environment for cell migration and ultimately to decreased numbers of WFs in mutants. Similar results were found by *Hudacova et al., 2025*, in craniofacial mesenchymal cells in *Meis2* cKO mice. These increases in cell adhesion and ECM gene expression might also contribute to decreased TG axonal growth and branching described above (*Figure 1*, *Videos 1 and 2*).

## *Meis2* controls expression of *Foxd1* in the dermis

Next, we performed differential gene expression analysis between the DC cluster of *Meis2* cKO and controls at E12.5 and E13.5. In this scRNA-seq dataset, *Foxd1* was found within the top genes downregulated in *Meis2* cKO mice (avg_log2FC = 1.0585, p_val = 4.55E-06) (*Figure 6A*, *Supplementary*

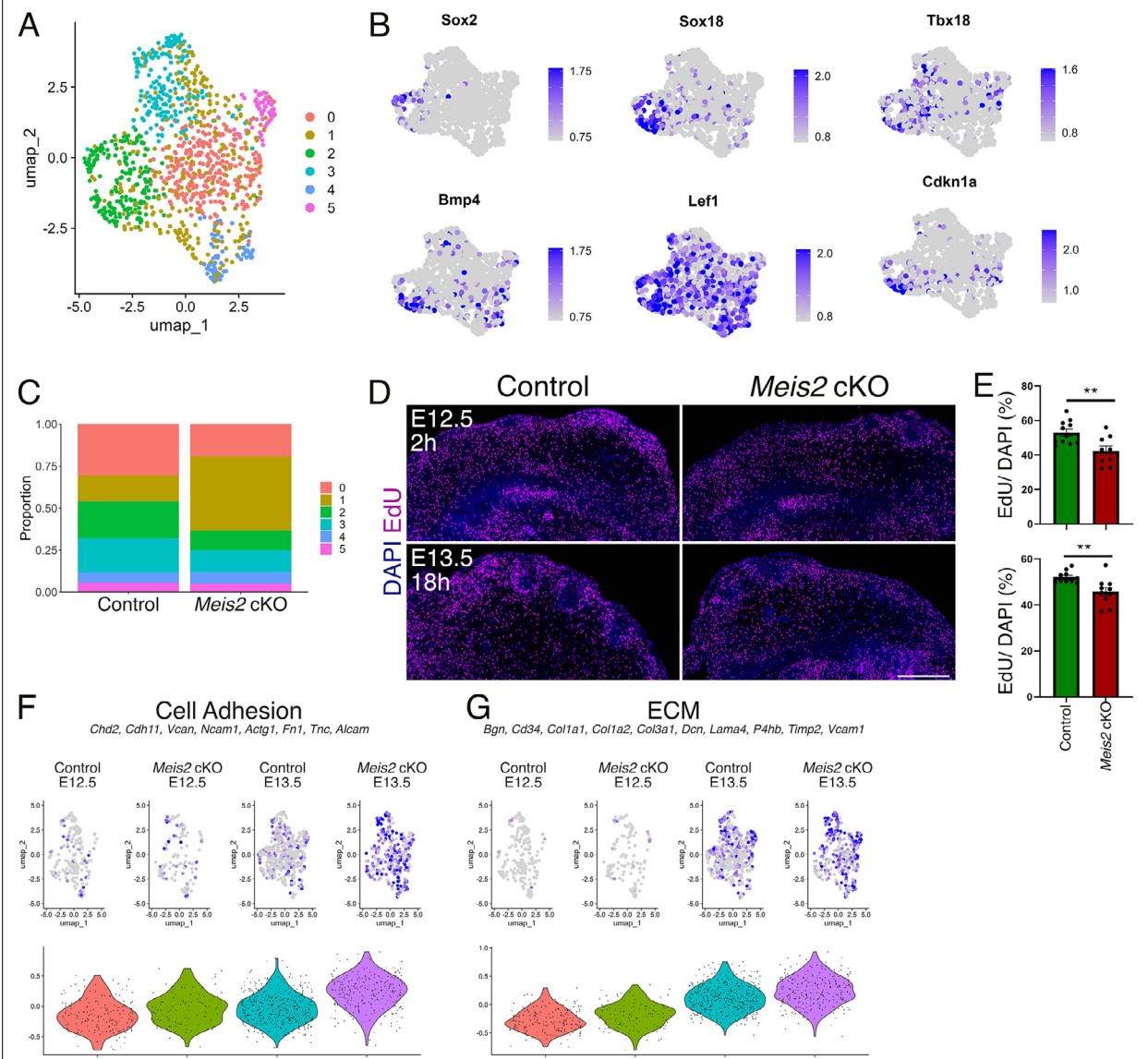

**Figure 5.** Dermal condensation and proliferation are reduced in *Meis2* cKO. (**A**) Uniform Manifold Approximation and Projection (UMAP) diagram of single-cell RNA-seq (scRNA-seq) analysis of the dermal mesenchyme subset with six-cluster resolution in which cluster 2 (green) represents dermal condensate (DC) of whisker follicles (WFs). (**B**) UMAP representations of typical DC markers *Sox2*, *Sox18*, *Tbx18*, *Bmp4*, *Lef1*, and *Cdkn1* shown by FeaturePlots. (**C**) Relative cell numbers in six clusters showing a higher cell count in cluster 1 (nonspecialized cells) at the expense of clusters 0 (dividing cells) and 2 (DC). (**D**) Analysis of cell proliferation by EdU incorporation after a 2 hr pulse at embryonic day 12.5 (E12.5) and an 18 hr pulse at E13.5. (**E**) Overall proliferation rate was reduced in *Meis2* cKO from 52.96±2.05% to 42.38 ± 2.77% (mean ± sem, t-test p=0.0057) in 2 hr pulse (top) and from 52.17±0.81% to 45.76 ± 1.85% (mean ± sem, t-test p=0.0052) in 18 hr pulse (bottom). Each data point indicates the average value for one section. n=2 mice and at least nine sections. (**F**) Increase of cell adhesion module score in mutants at E13.5 by scRNA-seq analysis. Module scores are generated by indicated genes. (**G**) Increase of extracellular matrix (ECM) module score in mutants at E13.5 by scRNA-seq analysis. Module scores are generated by indicated genes.

The online version of this article includes the following figure supplement(s) for figure 5:

**Figure supplement 1.** Identification of dermal condensate (DC) cluster in single-cell RNA sequencing (scRNA-seq) datasets.

*file 3*). Interestingly, *Foxd1* expression was described as a hallmark of the earliest molecular changes in dermal Fb prior to DC formation and defined the pre-DC stage in HF (*Mok et al., 2019*). Further, *Sox2*, a well-known DC marker, was also strongly downregulated in *Meis2* cKO (*Figure 6A*). To validate transcriptomic data, we analyzed FOXD1 protein expression in tissue sections of wild-type snouts at E12.5 and E13.5. Similarly to HF, *Foxd1* expression preceded *Sox2* also in WF which occurred

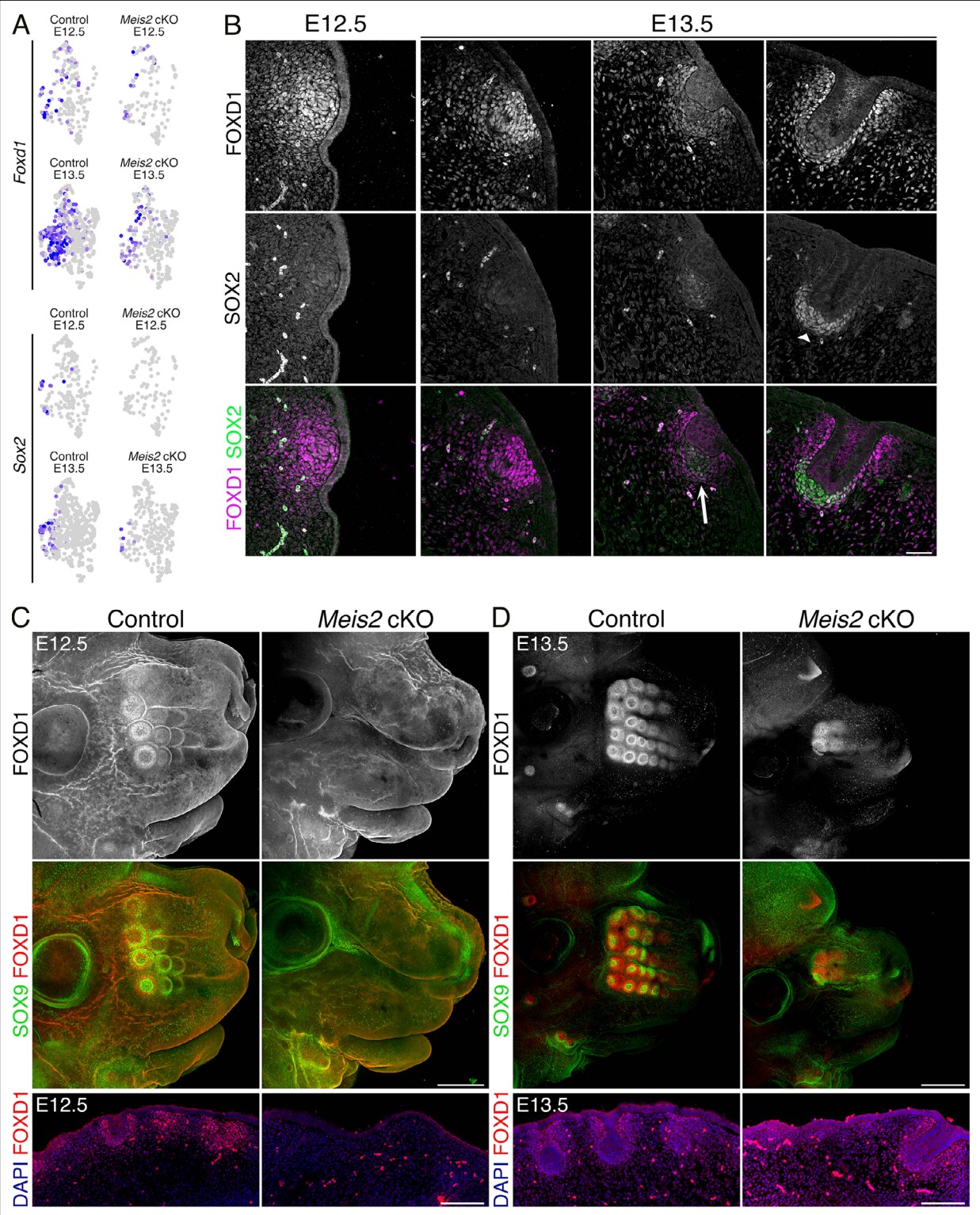

**Figure 6.** Expression of the earliest pre-dermal condensate (DC) and DC marker *Foxd1* is lost in *Meis2* cKO. (**A**) Uniform Manifold Approximation and Projection (UMAP) representations of decline in *Foxd1* (top) and *Sox2* (bottom) expression in DC cluster (#2) shown by FeaturePlots. Expression of both genes is significantly reduced in mutants. (**B**) Immunofluorescence of FOXD1 and SOX2 at initial stages of whisker follicle (WF) showing FOXD1 in the pre-DC stage while the expression shifts from DC to peri-DC regions during WF progression. SOX2 appears as a typical DC marker with no earlier expression. Scale bar: 50 μm. (**C–D**) Whole-mount immunostaining of FOXD1 and SOX9 of heads from controls and *Meis2* cKO at embryonic day 12.5 (E12.5) (**C**) and E13.5 (**D**). It shows the absence of WFs including the pre-DC marker FOXD1 at the E12.5 stage when proximal columns of WFs have

*Figure 6 continued on next page*

*Figure 6 continued*

already formed. A few WF escapers develop in mutants at E13.5 in proximal regions while at least six WF columns have formed in controls. Scale bars: 400 µm (**C**) and 500 µm (**D**). Lower panels: FOXD1 staining in FFPE sections. Scale bars: 150 µm.

before the Pc downgrowth into the dermal niche at E12.5. During initial Pc invagination and early DC stage at E13.5, *Foxd1* and *Sox2* expression overlapped (*Figure 6B*, arrows). In more progressed WF germs with deeply invaginated Pc, *Foxd1* expression remained in peri-DC regions rather than in DC that was marked by a strong SOX2 signal (*Figure 6B*, right panels, arrowhead). Next, whole-mount staining of FOXD1 in the whole snout highlighted dynamic WF development along the proximo-distal axis. It started in proximal columns at E12.5, in which WF formation was more progressed, while distal snout regions initiate WF slightly later (*Figure 6C*). By E13.5, three to four proximal columns displayed discernible WFs (*Figure 6D*). Strikingly, *Meis2* cKO snouts displayed no FOXD1+ WFs at E12.5 (*Figure 6C*). At E13.5, we detected around four proximal WFs in mutants while controls contained around 17 WFs on each snout side (*Figure 6D*). FOXD1+ WFs in *Meis2* cKO most likely represented 'escaper' WFs that were described above. Whole-mount FOXD1 immunostaining was further confirmed on tissue sections (*Figure 6C and D*, lower panels). Normal WF at E13.5 expressed *Foxd1* in DC or peri-DC and *Sox9* in Pc. In contrast, this progression of WF formation was halted in *Meis2* cKO (*Figure 6C and D*). These results suggest that MEIS2 is the key component of transcription controlling machinery in the dermal mesenchymal that initiates WF.

### *Foxd1* is dispensable for WF development

Because *Foxd1* expression in DC is strictly regionally specific and its expression is lost in *Meis2* cKO, we hypothesized that *Foxd1* transcription factor function is essential for WF formation. We therefore analyzed whiskers in embryos where *Foxd1* expression is ablated. We crossed heterozygous *Foxd1*-GFP-CreERT2 mice, in which GFP-Cre-ERT2 cassette was inserted into the initiation codon of *Foxd1* gene. Homozygous embryos are therefore null mutants for *Foxd1* gene. WF development was assessed in whole-mount-stained embryonic heads at E13.5 using SOX9, FOXD1, and TUJ1 antibodies. *Figure 7A* shows that SOX9 was detected in invaginating Pc, FOXD1 in surrounding DC mesenchyme, and TUJ1 in axonal projections innervating each WF. In parallel, we stained EDAR in WF Pc (*Figure 7B*). Interestingly, the number and morphology of WF in controls and *Foxd1* mutants

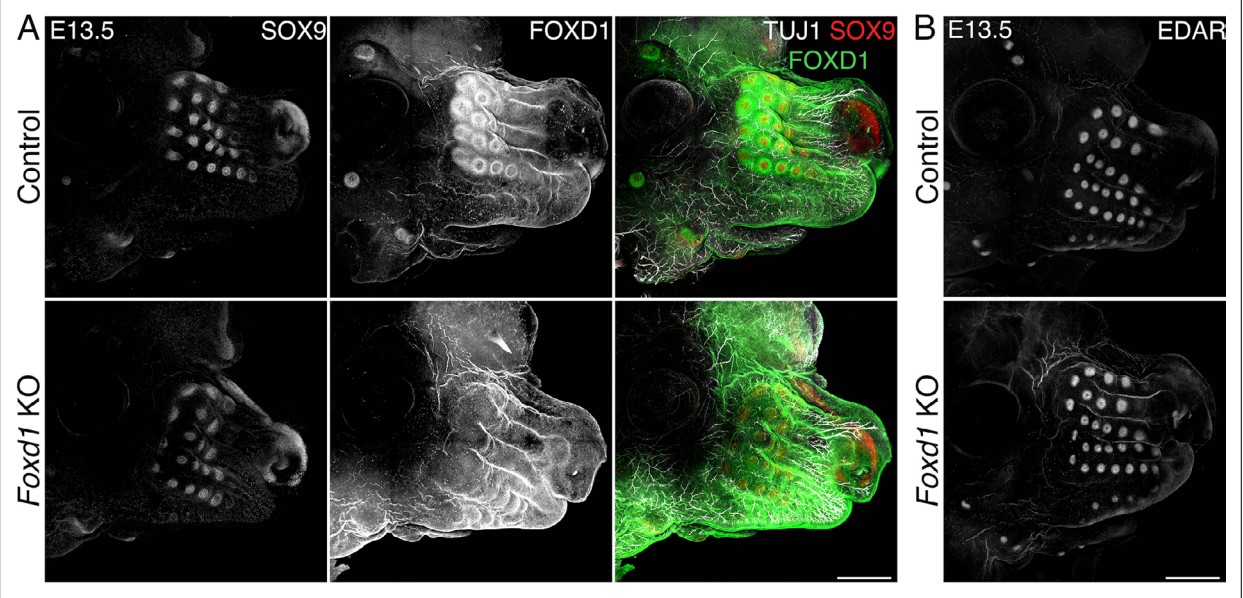

**Figure 7.** Whisker follicles (WFs) normally develop in loss-of-function *Foxd1* mutants. (**A**) Whole-mount immunostaining of FOXD1, TUJ1, and SOX9 of heads from controls and *Foxd1*-null mutants at embryonic day 13.5 (E13.5) showing normal formation of WFs in mutants in which FOXD1 signal disappears. Normal WF development is also reflected in normal WF innervation represented by TUJ1 staining. Scale bars: 500 µm. (**B**) Whole-mount immunostaining of EDAR confirmed normal placode (Pc) appearance in *Foxd1*-null mutants. Scale bars: 500 µm.

were essentially identical. Thus, FOXD1 protein is a very important marker of the early DC cell fate in WF but does not have functional relevance for whisker formation despite its dynamic and regionally specific expression pattern around the developing WFs.

## Discussion

We have shown that mesenchymal expression of transcription factor *Meis2* is essential for the initial steps of mouse whisker development. Mouse embryos, in which *Meis2* was deleted in tissues derived from neural crest cells by Wnt1-Cre2 driver, lost the majority of whiskers. The aberrant whisker phenotype in the mutants is accompanied by markedly reduced TG nerve branching in the snout. This suggests a novel function of *Meis2* in TG axonal growth, branching, and guidance. It may also implicate that whisker formation, or WF positional determination, may depend on early sensory innervation. We report here, however, that whisker formation is entirely independent of sensory innervation. *Neurog1*$^{-/-}$ mice completely lack TG nerves, including their ganglions, but whisker development is not affected. Moreover, analysis of *Neurog1*$^{-/-}$ mice at E17.5 reveals that sensory innervation is not required for the further growth and stabilization of WFs. The effect of TG innervation deficiency on postnatal WF development could not be tested because *Neurog1*$^{-/-}$ mice are lethal within a day after birth. All these experiments exclude the possibility that the whisker phenotype in *Meis2* cKO embryos is an indirect consequence of decreased TG nerve branching. It rather indicates that mesenchymal *Meis2* is a pivotal regulator of WF formation.

Our observations suggest that nerve branching phenotype in the mutant is independent of TG ganglia development or missing whiskers, as *Meis2* cKO mice showed normal TG ganglia size and nerve exit from the ganglia as well as defective TG nerve branches that do not innervate whiskers. Thus, *Meis2* rather influences TG branching and guidance through the snout. Description of the exact nature of factors controlling TG nerve branching downstream of MEIS2 warrants further investigation. A number of molecular pathways, such as SLIT/ROBO, SEMA/NRP/PLXN (*Taniguchi et al., 1997*; *Schwarz et al., 2008a*; *Schwarz et al., 2008b*; *Ulupinar et al., 1999*; *Cheng et al., 2001*), and EPHRIN/EPH (*North et al., 2010*) signaling, have been implicated during TG nerve development, and *Meis2* might act upstream of mesenchymal expression of ligands and/or TG expression of receptors of these pathways to control TG nerve development. One possible mechanism might involve SEMA3A, an inhibitory factor for TG nerve growth during tooth innervation, and *Sema3a* null mice display hyperbranched TG nerves (*Ulupinar et al., 1999*; *Dent et al., 2004*; *Moe et al., 2012*). Our scRNA-seq datasets indicated an increase of *Sema3a* expression in *Meis2* cKO mesenchymal cell populations (data not shown), which could explain decreased nerve branching in *Meis2* cKO mice and could be addressed in future studies.

Essential role of *Meis2* in the dermal mesenchyme during very early steps of WF development is supported by several key observations. First, although epithelial expression of *Meis2* is preserved in *Meis2* cKO mice, thickening of the epithelium is rarely observed, suggesting that the release of WF Pc-inducing dermal factors is under control of mesenchymal *Meis2*. It is widely accepted that HF or WF formation is initiated from the dermis by a 'first dermal signal' that has not been identified yet. Our data show that this process is dependent on *Meis2*. Second, the absence of *Meis2* results in the loss of early placodal markers LEF1 and EDAR and pre-DC marker FOXD1 at expected WF locations. Absence of patterned upregulation of LEF1 and EDAR was documented in HFs after blocking dermal WNT signaling (*Chen et al., 2012*). Interestingly, our analysis of WNT signaling readouts did now show remarkable changes in canonical WNT signaling in *Meis2* cKO dermis, indicating that normal dermal WNT signaling is not sufficient to trigger patterned LEF1 and EDAR upregulation in the epithelium when dermal MEIS2 is absent. These epithelial changes probably require additional *Meis2*-dependent intermediate steps, which might act either in parallel with or downstream of dermal WNT signaling to induce Pc formation. In the dermis, pre-DC marker FOXD1 expression was shown to be dependent on *Fgf20* expression in HF Pc and occurs after Pc formation (*Mok et al., 2019*). Our analysis of *Foxd1* expression in wild-type snout dermis revealed that *Foxd1* expression appears prior to morphologically distinguishable WF Pc. In *Meis2* cKO, FOXD1 signal in the dermis prior to WF Pc formation was not detected, which accords with scRNA-seq data. This all suggests that *Meis2* operates upstream of *Foxd1*. However, direct analysis of *Foxd1*-null mice did not reveal any phenotype in WF development. This could be explained by either no function of *Foxd1* in this process or by a possible functional compensation by its paralog *Foxd3*. Since *Foxd1* does not affect WF formation and therefore the

absence of *Foxd1* cannot account for lacking WFs in *Meis2* cKO mice, loss of FOXD1 in these mice evidently is a result of lacking pre-DC and DC structures. Lastly, it has been shown in HF development that dermal Fb proliferate extensively prior to converting to DC cell fate. DC cells, on the other hand, exit the cell cycle and become quiescent (*Mok et al., 2019*; *Gupta et al., 2019*). We generally observed reduced proliferation in upper dermal cells in *Meis2* cKO mice. Lower cell proliferation could contribute to inefficient cell fate change of Fb to DC and might also lead to insufficient expression and/or release of Pc-inducing factor in the dermis of *Meis2* mutants. Overall, these observations strongly suggest that the absence of *Meis2* in mesenchymal cells affects earliest processes of WF development, even prior to Pc formation.

It is important to note that we observed small numbers (around 18% at E13.5) of WFs in *Meis2* cKO expressing normal placodal and DC markers. These rarely formed WF 'escapers' in *Meis2* cKO mice had normal WF morphology, because their Pc thickness, follicle length, and DC size were comparable between controls and mutants at E12.5-E13.5. WF 'escapers' in mutants were probably not sensitive to Wnt1-Cre2-mediated deletion of *Meis2*. Although immunostaining of MEIS2 in snout sections indicated that *Meis2* expression around WF escapers was hardly detectable in mutants, a slight but undetectable expression of *Meis2* resulting from incomplete gene deletion might account for normal WF formation. On the other hand, it is also possible that *Meis1* paralog can partially substitute for missing *Meis2* in its function in the dermal mesenchyme. Alternatively, *Meis2* function in WF development might be restricted to a short time window as each WF develops at different time points along the proximal-distal axis. It also remains as a tentative possibility that *Meis2* in the whisker pad mesenchyme is required for yet an unidentified function. Therefore, the absence of whiskers could be an indirect consequence of this plausibility.

## Materials and methods
### Mouse strains
Generation and genotyping of the floxed allele of *Meis2* gene (*Meis2* $^{fl/fl}$) was described by *Machon et al., 2015*. *Meis2*$^{fl/fl}$ strain was crossed with reporter mouse lines R26R-*EYFP* $^{fl/fl}$ (RRID:IMSR_JAX:006148) before crossing with the Wnt1-Cre2 (*Lewis et al., 2013*). This reporter enabled monitoring Cre recombination specificity. Wnt1-Cre2 (RRID:IMSR_JAX:022137) was used for specific deletion of the *Meis2*$^{fl/fl}$ gene in neural crest cells. For lineage tracing experiments, we used Wnt1-Cre2 crossed with mTmG (RRID:IMSR_JAX:007676). During embryo harvesting, Cre recombination was checked by monitoring EYFP fluorescence. In very rare cases (1–2 embryos in 10 litters), Cre activity in the whole body was observed, indicating germline recombination. Such animals were removed from the analysis, and their parents were removed from crossing schemes. Neurogenin1 knockout mice were purchased from Jax Lab (RRID:IMSR_JAX:017306). *Foxd1*-GFP-Cre-ERT2 mice were purchased from Jax Lab (RRID:IMSR_JAX:012464). The presence of the vaginal plug was regarded as the E0.5. Pregnant mice were anesthetized using isoflurane gas inhalation and euthanized by cervical dislocation. Embryos were dissected and placed in ice-cold PBS. All mice were genotyped as previously described (*Machon et al., 2015*). *Neurog1* and *Foxd1* mutant mice were genotyped according to the Jax protocol. All procedures involving experimental animals were approved by the Institutional Committee for Animal Care and Use (permission #PP-10/2019) with every effort to minimize suffering and the number of animals used. This work did not include human subjects.

### Immunohistochemistry and image processing
All embryos used for staining were harvested at either E12.5 or E13.5 embryonic stages, and heads were fixed in 4% paraformaldehyde (PFA) overnight at 4°C. After PFA fixation, samples were prepared in three ways. (1) For FFPE staining, after dehydration in ethanol and xylene, samples were embedded in paraffin, and frontal 10 µm sections were prepared. Followed by stepwise rehydration, antigen retrieval in 0.1 M citrate buffer, pH 6.0, under pressure boiling for 12 min was carried out. (2) For cryosection staining, after dehydration in 30% saccharose for 24 hr, samples were embedded in OCT, and 100 µm cryosections were obtained. (3) For whole-mount staining, whole heads were directly processed to permeabilization and blocking after PFA fixation.

Tissues were blocked in 5% bovine serum albumin (BSA) in PBS with 0.1% Triton X-100 for FFPE sections or with 0.5% Triton X-100 for thick cryosections or whole mounts for 1 hr and incubated in

primary antibody solution (1% BSA in PBS and 0.1% Triton X-100 for 10 µm sections or 0.5% Triton X-100 for 100 µm sections and whole mounts). Primary antibody incubation was overnight for FFPE sections, two nights for cryosections, and three nights for whole-mount staining at 4°C. Primary antibodies used: SOX9 (Merck Sigma, AB5535), TRKA (R&D Systems, AF1056), MEIS2 (GeneScript, custom), TUJ1 (R&D Systems, MAB1195), EDAR (R&D Systems, AF745), Lef1 (Cell Signaling, C12A5), beta-galactosidase (Abcam, ab9361), SOX2 (R&D Systems, MAB2018), FOXD1 (Abcam, AB129324). Primary antibody solutions were washed out with PBS, Triton X-100 and incubated in fluorescent secondary antibody solutions for 1 hr for FFPE sections, 3 hr for cryosections at room temperature or overnight for whole mounts at 4°C. Secondary antibodies: donkey anti-mouse, -rabbit, -goat with Alexa Fluor 488, 568, or 647 fluorophores (Thermo Fisher Scientific). Primary and secondary antibodies were used at 1:500 for thick section and whole-mount staining and at 1:1000 for thin section staining. Immunofluorescent images were scanned by spinning disc confocal microscopy using an Olympus SpinSR10. Imaging was performed for whole-mount stainings after tissue clearing for 1 hr at room temperature. Acquired z-stacks were maximum intensity projected using ImageJ. Figures were assembled using Photoshop.

For cell proliferation assays, EdU was injected intraperitoneally to pregnant females at E12.5 for 2 or 18 hr before harvesting (0.5 ml per mouse, 2.5 mg/ml). EdU Base Click kit was used for EdU detection according to the manufacturer's protocol on FFPE sections.

## Micro-CT

Embryos were fixed in 4% PFA for 2 days and soaked in Lugol's iodine for several days. Scanning was performed on the instrument Bruker Skyscan 1272 with the resolution 3 µm.

## In situ hybridization

Antisense RNA probe for *Shh* was cloned into pGEM-T-easy vector (Promega) using primers: *Shh*-F TCACAAGTCCTCAGGTTCCG, *Shh*-R GGGCTTCAGCTGGACTTGAC. Antisense mRNA was transcribed with T7 polymerase. Whole-mount in situ hybridization was performed using standard protocols (*Machon et al., 2015*). Fluorescent in situ hybridization HCR RNA-FISH (Molecular Instruments) was performed according to the manufacturer's protocols.

## scRNA-seq analysis

scRNA-seq datasets are retrieved from integrated Seurat objects (GSE262468) (*Hudacova et al., 2025*) and processed with the standard Seurat workflow (*Hao et al., 2021*). Briefly, dermal Fb clusters were subset and split by condition, which was followed by SCTransform normalization, where mitochondrial, ribosomal, and cell cycle genes were regressed. Seurat objects were then integrated by using the IntegrateData function with 3000 integration features. For UMAP as a nonlinear dimensional reduction embedding, 30 principal components were used. We used the Clustree package to determine the optimal resolution, which was set at 0.6 for the FindClusters function according to the developers' instructions (*Zappia and Oshlack, 2018*). SCT assay was used to generate feature plots and violin plots. Normalized and Scaled data of the RNA assay were used to extract cluster markers using the FindAllMarkers function and to generate heatmaps. The FindMarkers function of the Seurat package was used for differential gene expression analysis between conditions with default parameters. clusterProfiler (4.6.2) was used for GO analysis (*Wu et al., 2021*), where GO terms were extracted by the enrichGO function. This paper did not produce original codes. Codes used for analysis and lists of genes used to generate module scores can be accessed at https://github.com/kaplanmm/whisker_scRNA (copy archived at *Kaplan, 2025*).

## Quantification and statistical analysis

For Pc thickness/WF length and DC size analysis, images of DAPI-stained 10 µm FFPE snout sections were captured with an Olympus SpinSR10 microscope and a ×20 objective. With ImageJ software, maximum intensity projection was applied to z-stack images. DAPI staining was used to visually identify Pc and DCs. The distance from the outer layer of epithelium in the placodal region to the epithelial-mesenchymal boundary was measured. For 2D DC size, DCs were identified by densely accumulated cells underneath the Pc. The area of the region covered by these cells was measured using ImageJ.

The number of WFs was manually counted in embryos that were used throughout the study for whole-mount immunostainings (labeled by SOX9, EDAR, and/or FOXD1 antibodies) or in situ hybridization (*Shh* mRNA). The numbers of WFs at both sides of the snout were pooled. The decrease in the percentage of WFs in *Meis2* cKO mice was calculated in comparison to their control littermates. Values from each experiment were averaged to determine the overall decrease in the percentage of WFs.

For Lef1 quantification, images of Lef1-stained 10 µm FFPE snout sections were captured with an Olympus SpinSR10 microscope and a ×20 objective. With ImageJ software, maximum intensity projection was applied to z-stack images. Local thickenings of the epithelium, Pc, were determined visually by DAPI staining. Their Lef1 positivity was determined and manually counted. Lef1 fluorescence intensity of these sites were measured by using ImageJ intensity analysis tool after manually drawing areas covering Lef1+ Pc. Similarly, Lef1 fluorescence intensity of DCs was measured with ImageJ after drawing areas covering DCs that were densely populated with DAPI-stained cells underneath the Pc. Lef1 fluorescence intensities in non-DC upper dermal regions were measured in the dermal region between the epithelium and around a 100 µm distance into the dermis between DCs.

For EdU/DAPI area measurements, images of EdU-chased and DAPI-stained 10 µm FFPE snout sections were captured with an Olympus SpinSR10 microscope equipped with a ×20 objective. With ImageJ software, maximum intensity projection was applied to z-stack images. The dermal region between the epithelium and around a 100 µm distance into the dermis was used for analysis, while DC domains were excluded. The selected dermal regions were processed separately for thresholding steps to cover EdU+ and DAPI+ regions. Threshold values were determined visually to cover specific EdU and DAPI staining. The same threshold values were used for control and *Meis2* cKO mutants. Total area values for EdU and DAPI were used to calculate 2D EdU coverage of the DAPI regions (EdU/DAPI).

Statistical analyses were performed using GraphPad Prism 10 software. N numbers specified in the respective figure legends for each analysis. All the data are presented with mean ± sem (standard error of the mean). Two-tailed Student's t-test was used to explore statistical differences between two experimental groups (control vs. *Meis2* cKO).

## Acknowledgements

This work was supported by the Czech Science Foundation (grants 22-10660S, 23-06160S) and Masaryk University, Faculty of Medicine (MUNI/A/1598/2023). We thank the Microscopy Service Centre of IEM CAS and Light Microscopy Core Facility, IMG CAS, Prague, Czech Republic, supported by MEYS – LM2023050 and RVO – 68378050-KAV-NPUI. This work was also funded by MEYS Czech Republic (NanoEnviCZ, LM2018124) and EU Structural Funds Pro-NanoEnviCz (CZ.02.1.01/0.0/0.0/16_013/0001821).

## Additional information

### Funding

| Funder | Grant reference number | Author |
|---|---|---|
| Czech Science Foundation | 22-10660S | Mehmet Mahsum Kaplan<br>Erika Hudacova<br>Miroslav Matejcek<br>Ondrej Machon |
| Masaryk University | MUNI/A/1598/2023 | Haneen Tuaima<br>Jan Křivánek |
| MEYS Czech Republic | NanoEnviCZ | Ondrej Machon |
| MEYS Czech Republic | LM2018124 | Ondrej Machon |
| EU Structural Funds Pro-NanoEnviCz | CZ.02.1.01/0.0/0.0/16_013/0001821 | Ondrej Machon |
| Czech Science Foundation | 23-06160S | Jan Křivánek |

| Funder | Grant reference number | Author |
|---|---|---|

The funders had no role in study design, data collection and interpretation, or the decision to submit the work for publication.

## Author contributions

Mehmet Mahsum Kaplan, Conceptualization, Data curation, Validation, Investigation, Visualization, Methodology, Writing – original draft, Writing – review and editing; Erika Hudacova, Miroslav Matejcek, Data curation, Methodology; Haneen Tuaima, Investigation, Methodology; Jan Křivánek, Resources; Ondrej Machon, Conceptualization, Data curation, Supervision, Funding acquisition, Project administration, Writing – review and editing

## Author ORCIDs

Mehmet Mahsum Kaplan ⓘ https://orcid.org/0000-0002-3013-0848
Jan Křivánek ⓘ https://orcid.org/0000-0002-7590-187X
Ondrej Machon ⓘ https://orcid.org/0000-0002-5139-1406

## Ethics

All procedures involving experimental animals were approved by the Institutional Committee for Animal Care and Use (permission #PP-10/2019) with every effort to minimize suffering and the number of animals used.

Reviewer #1 (Public review): https://doi.org/10.7554/eLife.100854.3.sa1
Reviewer #2 (Public review): https://doi.org/10.7554/eLife.100854.3.sa2
Author response https://doi.org/10.7554/eLife.100854.3.sa3

# Additional files

## Supplementary files

Supplementary file 1. Seurat cluster markers of dermal fibroblast. List of markers of all identified clusters for integrated samples of dermal fibroblasts from embryonic day 12.5 (E12.5) and E13.5 *Meis2* cKO mice and their control littermates extracted by FindAllMarkers Seurat function. Dermal fibroblast cell cluster is subset using single-cell RNA sequencing (scRNA-seq) data by Hudacova et al., 2025.

Supplementary file 2. clusterProfiler gene ontology (GO) for cluster 0 markers. clusterProfiler GO in biological processes run by using top 100 markers of cluster 0 of dermal fibroblast cell cluster from embryonic day 12.5 (E12.5) and E13.5 *Meis2* cKO and their control litters. GO terms were extracted by enrichGO function with default parameters.

Supplementary file 3. DEGs between dermal condensate (DC) cluster (cluster 2) of control and *Meis2* cKO mice. DEGs between DC cluster of integrated embryonic day 12.5 (E12.5) and E13.5 *Meis2* cKO mice and their control littermates, extracted by FindMarkers Seurat function where ident.1 and ident.2 arguments were set to control cluster 2 and *Meis2* cKO cluster 2, respectively.

MDAR checklist

## Data availability

Our previously published sequencing data used in this study were deposited to GEO under accession codes GSE262468.

The following previously published dataset was used:

| Author(s) | Year | Dataset title | Dataset URL | Database and Identifier |
|---|---|---|---|---|
| Hudacova E, Abaffy P, Kaplan MM, Krausova M, Kubista M, Machon O | 2025 | Gene expression profile of FACS sorted cranial neural crest cells from E12.5-E13.5 mouse embryonic head | https://www.ncbi.nlm.nih.gov/geo/query/acc.cgi?acc=GSE262468 | NCBI Gene Expression Omnibus, GSE262468 |

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
