## [Editor Report · eLife Assessment]

This study provides **valuable** insight into the role of Meis2 in whisker hair follicle formation and confirms prior work that nerves are dispensable for this process. The **solid** imaging techniques support the authors' conclusions, however the data provides limited evidence to support the mechanism of Meis2 in whisker formation.

---

## [Referee Report · Reviewer #1 (Public review)]

Summary:

Mehmet Mahsum Kaplan et al. demonstrate that Meis2 expression in neural crest-derived mesenchymal cells is crucial for whisker follicle (WF) development, as WF fails to develop in wnt1-Cre;Meis2 cKO mice. Advanced imaging techniques effectively support the idea that Meis2 is essential for proper WF development and that nerves, while affected in Meis2 cKO, are dispensable for WF development and not the primary cause of WF developmental failure. The study also reveals that although Meis2 significantly downregulates Foxd1 in the mesenchyme, this is not the main reason for WF development failure. The paper presents valuable data on the role of mesenchymal Meis2 in WF development. However, it is still not known what is the molecular mechanisms that link Meis2 to impact the epithelial compartment.

Strengths:

(1) The authors describe a novel molecular mechanism involving Mesenchymal Meis2 expression, which plays a crucial role in early WF development.

(2) They employ multiple advanced imaging techniques to illustrate their findings beautifully.

(3) The study clearly shows that nerves are not essential for WF development.

Weaknesses:

The paper lacks clarity on how Meis2 loss, along with the observed general reduction in proliferation and changes in extracellular matrix and cell adhesion, leads specifically to the loss of whisker follicles. Future studies addressing this gap, perhaps with methods enabling higher cell recovery or epithelial cell inclusion in the sequenced cells, could provide valuable insights into the specific roles of Meis2 in this context.

---

## [Referee Report · Reviewer #2 (Public review)]

Summary:

In this manuscript, Kaplan et al. study mesenchymal Meis2 in whisker formation and the links between whisker formation and sensory innervation. To this end, they used conditional deletion of Meis2 using the Wnt1 driver. Whisker development was arrested at the placode induction stage in Meis2 conditional knockouts leading to absence of expression of placodal genes such as Edar, Lef1, and Shh. The authors also show that branching of trigeminal nerves innervating whisker follicles was severely affected but that whiskers did form in the complete absence of trigeminal nerves.

Strengths:

The analysis of Meis2 conditional knockouts shows convincingly lack of whisker formation and all epithelial whisker/hair placode markers analyzed. Using Neurog1 knockout mice, the authors show that whiskers and teeth develop in the complete absence of trigeminal nerves.

Comments on revised version:

In the revised manuscript, Kaplan et al. have addressed some of my previous concerns, e.g., the methodological section has been updated to include the relevant information, and the Introduction now better considers the previous literature.

In the revised manuscript, the authors have made limited efforts to address the main criticism of my original review: lack of mechanistic insight as to why mesenchymal Meis2 leads to the absence of whisker placodes. The new data reported indicate that the lack of whisker placodes is not a mere delay. In this context, the authors also show one images of E18.5 snouts that includes developing hair follicles. Interestingly, the image shown seems to indicate that hair follicles do develop normally in the absence of mesenchymal Meis2 although this finding is not reported in any detail or quantified. The authors suggest that this could be due to an early role of Meis2 in the mesenchyme because HFs develop later. Indeed, one plausible possibility is that Meis2 does not have any direct role in whisker (or hair) follicle development but is specifically required for some other function in the whisker pad mesenchyme, a function that remains unidentified in the current study as it mainly focuses on analyzing hair follicle marker expression in whisker follicles. I think this should be better reflected in the Discussion.

Additional comments:

The revised manuscript included the quantification of Lef1 intensity in control and Meis2 cKO whisker follicles (lines 251-252 and 255-258). Maybe I missed, but I failed to find the information how the quantification of the intensities was made, and therefore it was not possible for me to evaluate this part of the data. Nevertheless, I think the main text is not the place for these quantifications; rather, they would better fit e.g. Suppl. Figure 4.

---

## [Author Response]

The following is the authors’ response to the original reviews.

**Public Reviews:**

**Reviewer #1 (Public review):**
Summary:Mehmet Mahsum Kaplan et al. demonstrate that Meis2 expression in neural crest-derived mesenchymal cells is crucial for whisker follicle (WF) development, as WF fails to develop in wnt1-Cre;Meis2 cKO mice. Advanced imaging techniques effectively support the idea that Meis2 is essential for proper WF development and that nerves, while affected in Meis2 cKO, are dispensable for WF development and not the primary cause of WF developmental failure. The study also reveals that although Meis2 significantly downregulates Foxd1 in the mesenchyme, this is not the main reason for WF development failure. The paper presents valuable data on the role of mesenchymal Meis2 in WF development. However, further quantification and analysis of the WF developmental phenotype would be beneficial in strengthening the claim that Meis2 controls early WF development rather than causing a delay or arrest in development. A deeper sequencing data analysis could also help link Meis2 to its downstream targets that directly impact the epithelial compartment.Strengths:(1) The authors describe a novel molecular mechanism involving Mesenchymal Meis2 expression, which plays a crucial role in early WF development.(2) They employ multiple advanced imaging techniques to illustrate their findings beautifully.(3) The study clearly shows that nerves are not essential for WF development.

We thank the reviewer for valuable comments that will help improve our study.

Weaknesses:(1) The authors claim that Meis2 acts very early during development, as evidenced by a significant reduction in EDAR expression, one of the earliest markers of placode development. While EDAR is indeed absent from the lower panel in Figure 3C of the Meis2 cKO, multiple placodes still express EDAR in the upper two panels of the Meis2 cKO. The authors also present subsequent analysis at E13.3, showing one escaped follicle positive for SHH and Sox9 in Figures 1 and 3. Does this suggest that follicles are specified but fail to develop? Alternatively, could there be a delay in follicle formation? The increase in Foxd1 expression between E12.5 and E13.5 might also indicate delayed follicle development, or as the authors suggest, follicles that have escaped the phenotype. The paper would significantly benefit from robust quantification to accompany their visual data, specifically quantifying EDAR, Sox9, and Foxd1 at different developmental stages. Additionally, analyzing later developmental stages could help distinguish between a delay or arrest in WF development and a complete failure to specify placodes.

The earliest DC (FOXD1) and placodal (EDAR, LEF1) markers tested in this study were observed only in the escaped WFs whereas these markers were missing in expected WF sites in mutants. This was also reflected in the loss of typical placodal morphology in the mutant’s epithelium. On the other hand, escaped WFs developed normally as shown by the analysis in Supp Fig 1A-B showing their normal size. These data suggest that development of escaped WFs is not delayed because they would appear smaller in size. To strengthen this conclusion, we assessed whisker development at E18.5 in *Meis2* cKO mice by EDAR staining and results are shown in newly added Supplementary Figure 2. This experiment revealed that whisker phenotype persisted until E18.5 therefore this phenotype cannot be explained by a developmental delay.

As far as quantification is concerned, we have already quantified the number of whiskers in controls and mutants at E12.5 and E13.5 in all whole mount experiments we did, i.e. *Shh* ISH and SOX9 or EDAR whole mount IFC. We pooled all these numbers together and calculated the whisker number reduction to 5.7+/-2.0% at E12.5 and 17.1+/-5.9 at E13.5. Line:132-134.

(2) The authors show that single-cell sequencing reveals a reduction in the pre-DC population, reduced proliferation, and changes in cell adhesion and ECM. However, these changes appear to affect most mesenchymal cells, not just pre-DCs. Moreover, since E12.5 already contains WFs at different stages of development, as well as pre-DCs and DCs, it becomes challenging to connect these mesenchymal changes directly to WF development. Did the authors attempt to re-cluster only Cluster 2 to determine if a specific subpopulation is missing in Meis2 cKO? Alternatively, focusing on additional secreted molecules whose expression is disrupted across different clusters in Meis2 cKO could provide insights, especially since mesenchymal-epithelial communication is often mediated through secreted molecules. Did the authors include epithelial cells in the single-cell sequencing, can they look for changes in mesenchyme-epithelial cell interactions (Cell Chat) to indicate a possible mechanism?

We agree with the reviewer that the effect of Meis2 on cell proliferation and expression of cell adhesion and ECM markers are more general because they take place in the whole underlying mesenchyme. Our genetic tools did not allow specific targeting of DC or pre-DCs. Nonetheless, we trust that our data show that mesenchymal *Meis2* is required for the initial steps of WF development including Pc formation. As far as bioinformatics data are concerned, this data set was taken from the large dataset GSE262468 covering the whole craniofacial region which led to very limited cell numbers in the cluster 2 (DC): WT_E12_5  28, WT_E13_5  131, MUT_E12_5  19, MUT_E13_5  28. Unfortunately, such small cell numbers did not allow further sub-clustering, efficient normalization, integration and conclusions from their transcriptional profiles. Although a number of interesting differentially expressed genes were identified (see supplementary datasets), none of them convincingly pointed at reasonable secreted molecule candidate.

We agree with the reviewer that cellchat analysis could provide robust indication of the mesenchymal-epithelial communication, however our datasets included only mesenchymal cell population (Wnt1-Cre2progeny) and epithelial cells were excluded by FACS prior to sc RNA-seq. (Hudacova et al. https://doi.org/10.1016/j.bone.2024.117297)

(3) The authors aim to link Meis2 expression in the mesenchyme with epithelial Wnt signaling by analyzing Lef1, bat-gal, Axin1, and Wnt10b expression. However, the changes described in the figures are unclear, and the phenotype appears highly variable, making it difficult to establish a connection between Meis2 and Wnt signaling. For instance, some follicles and pre-condensates are Lef1 positive in Meis2 cKO. Including quantification or providing a clearer explanation could help clarify the relationship between mesenchymal Meis2 and Wnt signaling in both epidermal and mesenchymal cells. Did the authors include epithelial cells in the sequencing? Could they use single-cell analysis to demonstrate changes in Wnt signaling?

We have now analyzed changes in LEF1 staining intensity in the epithelium and in the upper dermis. According to these quantifications, we observed a considerable decline in the number of LEF1+ placodes in the epithelium which corresponds to the lower number of placodes. On the other hand, LEF1 intensity in the ‘escaped’ placodes were similar between controls and mutants. LEF1 signal in the upper dermis is very strong overall and its quantification did not reveal any changes in the DC and non-DC region of the upper dermis. These data corroborate with our conclusion that *Meis2* in the mesenchyme is not crucial for the dermal WNT signaling but is required for induction of LEF1 expression in the epithelium. However, once ‘escaper’ placodes appear, they display normal wnt signaling in Pc, DC and subsequent development. These quantitative data have been added to the revised manuscript. Line247-260.

(4) Existing literature, including studies on Neurog KO and NGF KO, as well as the references cited by the authors, suggest that nerves are unlikely to mediate WF development. While the authors conduct a thorough analysis of WF development in Neurog KO, further supporting this notion, this point may not be central to the current work. Additionally, the claim that Meis2 influences trigeminal nerve patterning requires further analysis and quantification for validation.

We agree with the reviewer that analysis of the *Neurogenin1* knockout mice should not be central to this report. Nonetheless, a thorough analysis of WF development in *Neurog1* KO was needed to distinguish between two possible mechanisms: whisker phenotype in *Meis2* cKO results from 1. impaired nerve branching 2. Function of *Meis2* in the mesenchyme. We will modify the text accordingly to make this clearer to readers. We also agree that nerve branching was not extensively analyzed in the current study but two samples from mutant mice were provided (Fig1 and Supp Videos), reflecting the consistency of the phenotype (see also Machon et al. 2015). This section was not central to this report either but led us to focus fully on the mesenchyme. We think that *Meis2* function in cranial nerve development is very interesting and deserves a separate study.

We have edited the introduction to reflect the literature better. Line70-79.

(5) Meis2 expression seems reduced but has not entirely disappeared from the mesenchyme. Can the authors provide quantification?

We have attempted to quantify MEIS2 staining in the snout dermis. However, the background fluorescence made it challenging to reliable quantify. Additionally, since at the point, dermal region where MEIS2 expression is relevant to induce WF formation is not known, we were unable to determine the regions to analyze. Instead, we now added three additional images from multiple regions of the snout sections stained with MEIS2 antibody in Supplementary Figure 1C. We believe newly added images will make our conclusion that MEIS2 is efficiently deleted in the mutants more convincing.

**Reviewer #2 (Public review):**
Summary:In this manuscript, Kaplan et al. study mesenchymal Meis2 in whisker formation and the links between whisker formation and sensory innervation. To this end, they used conditional deletion of Meis2 using the Wnt1 driver. Whisker development was arrested at the placode induction stage in Meis2 conditional knockouts leading to the absence of expression of placodal genes such as Edar, Lef1, and Shh. The authors also show that branching of trigeminal nerves innervating whisker follicles was severely affected but that whiskers did form in the complete absence of trigeminal nerves.Strengths:The analysis of Meis2 conditional knockouts convincingly shows a lack of whisker formation and all epithelial whisker/hair placode markers were analyzed. Using Neurog1 knockout mice, the authors show equally convincingly that whiskers and teeth develop in the complete absence of trigeminal nerves.

We thank the reviewer for valuable comments that will help improve our study.

Weaknesses:The manuscript does not provide much mechanistic insight as to why mesenchymal Meis2 leads to the absence of whisker placodes. Using a previously generated scRNA-seq dataset they show that two early markers of dermal condensates, Foxd1 and Sox2, are downregulated in Meis2 mutants. However, given that placodes and dermal condensates do not form in the mutants, this is not surprising and their absence in the mutants does not provide any direct link between Meis2 and Foxd1 or Sox2. (The absence of a structure evidently leads to the absence of its markers.)

We apologize for unclear explanation of our data. We meant that *Meis2* is functionally upstream of *Foxd1* because *Foxd1* is reduced upon Meis2 deletion. This means that during WF formation, *Meis2* operates before *Foxd1* induction and does not mean necessarily that *Meis2* directly controls expression of *Foxd1*. Yes, we agree with reviewer’s note that *Foxd1* and *Sox2*, as known DC markers, decline because the number of WF declines. We wanted to convince readers that *Meis2* operates very early in the GRN hierarchy during WF development. We also admit that we provide poor mechanistic insights into *Meis2* function as a transcription factor. We think that this weak point does not lower the value of the report showing indispensable role of *Meis2* in WFs.

**Recommendations for the authors:**

**Reviewer #1 (Recommendations for the authors):**
The text could benefit from editing.

We have proofread the text.

Some information is missing from the materials and methods section - a description of sequenced cells, the ISH protocol used, etc.

Methodological section has been updated and single-cell experiments were performed and described in detail by Hudacova et al. 2025 (https://doi.org/10.1016/j.bone.2024.117297). We have utilized these datasets for scRNA analysis which has been described sufficiently in the referred paper. Reference for standard in site protocol has been added.

**Reviewer #2 (Recommendations for the authors):**
In the Introduction of the paper, the authors raise the question on the role of innervation in whisker follicle induction "It has been speculated that early innervation plays a role in initiating WF formation (ref. 1)"...and..."this revives the previous speculations that axonal network may be involved in WF positioning". However, the authors forget to mention that Wrenn & Wessless, 1984 (reference 1 in the manuscript) made exactly the opposite conclusion and stated e.g. "Nerve trunks and branches are present in the maxillary process well before any sign of vibrissa formation. Because innervation is so widespread there appears to be no immediate temporal correlation between the outgrowth of a nerve branch to a site and the generation of a vibrissa there. Furthermore, at the time just prior to the formation of the first follicle rudiment, there is little or no nerve branching to the presumptive site of that first follicle while branches are found more dorsally where vibrissae will not form until later." Therefore, I find that referring to the paper by Wrenn & Wessells is somewhat misleading. Given that the whisker follicles develop in ex vivo cultured whisker pads further hints that innervation is unlikely to play a role in whisker follicle induction.

The Introduction also hints at the role of innervation in tooth induction but forgets to refer to the literature that shows exactly the opposite. Based on the evidence it rather appears that the developing tooth regulates the establishment of its own nerve supply, not that the nerves would regulate induction of tooth development.

in my opinion, the Introduction should be partially rewritten to better reflect the literature.

The introduction has been revised to better reflect the literature on the role of innervation on WF and tooth development. Line70-87.

The authors conclude that Meis2 is upstream of Foxd1, but the evidence is based on the lack of Foxd1 expression in Meis2 mutants. However, as whiskers do not form, evidently all markers are also absent. More direct evidence of Meis2 being upstream of Foxd1 (or Sox2) should be presented to consolidate the conclusions.

We have already reacted to this point above in the section Weaknesses. The text is now modified so that the interpretation is correct. Line: 407-409.

Other comments:Author contributions state that XX performed experiments but the author list does not include anyone with such initials.

This error has been corrected in revision.